# Immunotherapy: Recent Advances and Its Future as a Neoadjuvant, Adjuvant, and Primary Treatment in Colorectal Cancer

**DOI:** 10.3390/cells12020258

**Published:** 2023-01-08

**Authors:** Irene Yu, Anthony Dakwar, Kazuaki Takabe

**Affiliations:** 1Department of Surgical Oncology, Roswell Park Comprehensive Cancer Center, Buffalo, NY 14263, USA; 2Department of Surgery, Jacobs School of Medicine and Biomedical Sciences, State University of New York, Buffalo, NY 14214, USA; 3Department of Gastroenterological Surgery, Yokohama City University Graduate School of Medicine, Yokohama 236-0004, Japan; 4Department of Surgery, Niigata University Graduate School of Medical and Dental Sciences, Niigata 951-8510, Japan; 5Department of Breast Surgery and Oncology, Tokyo Medical University, Tokyo 160-8402, Japan; 6Department of Breast Surgery, Fukushima Medical University, Fukushima 960-1295, Japan

**Keywords:** colorectal cancer, immunotherapy, immunoexclusion, immune checkpoint inhibitor, adoptive cell transfer therapy, tumor vaccine

## Abstract

Immunotherapy in colorectal cancer (CRC) has made great strides within the past decade. Immune checkpoint inhibitors are a class of immunotherapy and have been shown to greatly improve patient outcomes in mismatch repair-deficient (dMMR) CRC. Now, they are part of the standard of care for this subset of CRC. Because of this, there has been a growing interest in the efficacy and timing of immunotherapy for other subsets of CRC, including locally advanced, metastatic, and microsatellite stable (MSS). In this review, we aim to examine the three main classes of immunotherapy for CRC—immune checkpoint inhibitors (ICIs), adoptive cell transfer therapy (ACT), and tumor vaccines—and discuss the most recent advances and future directions for each.

## 1. Introduction

Colorectal cancer (CRC) is often curable with surgery when discovered early. However, over 20% of cases present as metastatic disease, with an increasing number of cases being diagnosed in patients of younger age [1]. When the disease is more advanced, systemic therapy becomes the key to improving survival, which includes chemotherapy and, more recently, immunotherapy. Immunotherapy in CRC is still a fairly new treatment modality, but great strides have been made within the past decade. To date, there are three major classes of immunotherapy—immune checkpoint inhibitors, adoptive cell transfer therapy, and tumor vaccines—all of which act through different mechanisms. The greatest advances have been made with immune checkpoint inhibitors.

## 2. Immune Checkpoint Inhibitors

### 2.1. History of Immune Checkpoint Inhibitors

An immune checkpoint inhibitor (ICI) is a class of drug that prevents malignant cells from downregulating host cell immune responses against the tumor [2]. The first ICI that was approved for use is an antibody named Ipilimumab [3,4], which acts against cytotoxic T-lymphocyte-associated antigen 4 (CTLA-4), a ligand that allows tumors to evade the immune response. CTLA-4 is expressed on some T cells, such as regulatory T cells [5], which play a role in maintaining self-tolerance [6], as well as activated CD4^+^ T cells [7], which play a role in generating an antitumor response from the immune system [8]. The overall effect of CTLA-4 blockage is an increase in the immune system’s ability to recognize and attack cancer cells [9]. This was demonstrated to have a clinically significant effect first in metastatic melanoma.

The breakthrough for immunotherapy in CRC started with a landmark trial in 2012 that was sponsored by Bristol-Myers Squibb, the company that developed the anti-programmed cell death protein 1 (anti-PD-1) antibody Nivolumab. Tumor cells express programmed death-ligand 1 (PD-L1), the main ligand of PD-1, which is usually bound by activated T cells expressing the PD-1 receptor. The resulting signal cascade inhibits T-cell proliferation, weakens T-cell function [10], and decreases cytokine production [9], allowing the tumor cell to escape attack by the body’s immune system. Anti-PD-1 inhibitors block this signaling cascade, decreasing the tumor’s ability to evade the immune system. In the landmark trial, it was demonstrated that one in every four to five patients with melanoma, renal cell carcinoma, or non-small-cell lung cancer had significant responses to anti-PD-1 treatment [11]. There was no response in any of the CRC patients except for one patient with microsatellite instability (MSI). This patient had a complete response [12].

### 2.2. Relevance of Microsatellite Status in ICI Use

MSI is defined as genomic instability in cancer cells due to decreased length of microsatellites, which are repeated sequences of 1 to 6 base pairs, caused by a deficiency in DNA mismatch repair (dMMR) [13]. Approximately 15% of CRC is classified as MSI, which can be further divided into high microsatellite instability (MSI-H) and low microsatellite instability (MSI-L), where instability is demonstrated at greater than or less than 30–40% of loci, respectively [14]. The remaining 85% of CRC is classified as microsatellite stable (MSS), with proficiency in mismatch repair (pMMR) [15].

The mutations in these MSI tumors lead to high amounts of neoantigens, which are defined as peptides that have been altered by somatic mutations and are not expressed in normal, healthy tissues [16]. These neoantigens are particularly useful in immunotherapy when they are presented on major histocompatibility complex (MHC) molecules and when their expression induces an immune response by CD4^+^ or CD8^+^ T cells [17]. Such neoantigens then attract tumor-infiltrating lymphocytes (TILs), making the tumor “hot” and improving the response to immunotherapy [18,19]. In fact, this mechanism is not limited only to immune checkpoint inhibitors but is important across all types of immunotherapy.

This is nicely demonstrated in the NICHE trial, which evaluated the neoadjuvant response of early-stage colon cancer to the combination of Nivolumab and Ipilimumab [20]. Of the 21 dMMR and 20 pMMR tumors that were treated, 100% and 27% had a pathological response, respectively [20]. Because dMMR CRC showed such a robust response to Nivolumab, more studies began to evaluate the efficacy of other ICIs in this patient population. Pembrolizumab was one of the first two anti-PD-1 checkpoint inhibitors approved by the Food and Drug Administration for various cancers. It binds to a different epitope of PD-1 with a different affinity compared to Nivolumab [21]. In the Phase III KEYNOTE_177 study, Pembrolizumab was evaluated against chemotherapy in the setting of MSI [22]. A total of 307 patients with MSI-H dMMR CRC who had not received prior treatment for their cancer were enrolled in this study. They were randomized to a Pembrolizumab arm or a chemotherapy arm. Pembrolizumab was shown to be superior, both in progression-free survival and in adverse-effect profile. Further investigation into the quality of life of patients treated with Pembrolizumab also demonstrated significant improvements compared to chemotherapy [23]. Pembrolizumab and the combination of Nivolumab and Ipilimumab have now become options in the standard-of-care decision tree for MSI-H dMMR colon cancer according to the National Comprehensive Cancer Network (NCCN) Colon Cancer Guidelines [24].

While many mutations in the mismatch repair genes (MLH1, MSH2, MSH6, and PMS2) are sporadic, there are also germline mutations in mismatch repair genes, which are the defining mutations in Lynch syndrome. Lynch syndrome predisposes patients to a variety of malignancies, frequently of the colon and endometrium, but also of the ovaries, small bowel, and urinary tract [25]. Given the high response rate of MSI tumors to immunotherapy, the most recent version of the NCCN guidelines now recommends universal testing of patients with a personal history of CRC to determine whether mismatch repair or MSI mutations exist. This will help diagnose Lynch syndrome if present [24] and identify patients who would benefit significantly from Pembrolizumab. Interestingly, a recent systematic review suggested that there may not be a difference in response rates between sporadic MSI malignancies and malignancies in patients with Lynch Syndrome [25].

Because microsatellite status plays such a large role in determining the optimal treatment for patients presenting with CRC, strategies for determining microsatellite status are under active investigation. Generally, determination of microsatellite status occurs via two methods: (1) molecular assays of MSI and (2) immunohistochemistry of MMR proteins encoded by the MMR genes discussed above. Molecular assays use polymerase chain reaction to determine the instability of microsatellite markers and come in two panels: Bethesda and Pentaplex. The sensitivity of these panels ranges from 67 to 100% and specificity lies between 61 and 92% [26]. Immunohistochemistry testing classifies a tumor as dMMR if at least one MMR protein is determined to have complete nuclear loss of expression [26]. Interestingly, there have been past reports of discordance between immunohistochemistry testing and molecular assays, ranging from 1 to 10%, but a recent study demonstrated high concordance [26], suggesting that the newer tests are quite reliable and valid for clinical use. There are novel assessments of microsatellite status, such as the deep learning assay, which classifies tumors based on morphological patterns seen on pathology slides that are also under investigation [27].

Biomarkers predictive of good response to systemic therapy are being actively researched as well. One such biomarker is the tumor mutational burden (TMB), which quantifies the total number of somatic mutations per megabase of the tumor genome that has been interrogated [28]. Tumors with higher TMB have also been shown to have a higher neoantigen load, which, as previously discussed, increases a tumor’s response to immunotherapy. High TMB can be caused by dMMR mutations, but not all tumors with high TMB have an underlying germline mutation. In the KEYNOTE-158 trial, there were clinically meaningful responses to Pembrolizumab across patients with high TMB, but interestingly, MSI-H status did not account entirely for the clinical benefit seen [29]. Therefore, it is suggested that TMB assay testing occurs in tandem with germline mutation testing [30].

### 2.3. ICIs in dMMR Colon Cancer

Presently, ICIs are recommended for use only in MSI-H dMMR colon cancers under the following circumstances: locally unresectable or medically inoperable tumors, neoadjuvantly for clinical T4b tumors or resectable synchronous liver and/or lung metastases, as the primary treatment for unresectable synchronous tumors, and for recurrent or progressive tumors previously treated with resection and chemotherapy [24]. As demonstrated in the NICHE study, stage I-III dMMR colon cancer has a robust response to ICIs nearly 100% of the time, but in their cohort, only 60% had a complete pathologic response [20]. For stage I cancers, upfront resection has a high cure rate of greater than 90% without the need for additional adjuvant or neoadjuvant therapy [31]. This is important because while most dMMR tumors responded extremely well to ICIs, the use of ICIs is also associated with adverse events that can affect nearly every organ system. In fact, the risk of severe or life-threatening events ranged from 46 to 72% based on a systematic review and meta-analysis from 2018 by Xu et al. [32]. Toxicities more commonly associated with Pembrolizumab include arthralgias, hepatic toxicity, and pneumonitis, while Nivolumab affects the endocrine system, and Ipilimumab affects the skin, gastrointestinal tract, and kidneys [32]. Other commonly reported side effects of ICIs include myocarditis, vitiligo, adrenitis, myalgias and myositis, polyneuropathies, uveitis, interstitial lung disease, and vasculitis [33]. Because of this, it is likely that there will be limited use for ICIs in Stage I colon cancer.

For stage II and III cancers, however, the 10-year cumulative risk of recurrence is significantly higher [31,34]. The predicted cure rate is 88% and 71% for stage II and stage III cancers, respectively, while the actual observed cure for stage III cancer is only 63.6%, according to a recent prospective multicenter study by van den Berg et al. [31]. It is worth exploring the neoadjuvant role of ICIs for this patient population due to several benefits unique to neoadjuvant administration, which include pre-operative reduction in tumor size and evaluation of the efficacy of systemic treatment prior to resection. To this end, the NICHE-2 trial was conducted [35]. The cohort consisted of 112 dMMR CRC patients treated with one dose of Ipilimumab and two doses of Nivolumab. Of these, 95% had a major pathologic response and 67% demonstrated a complete pathologic response, similar to their initial trial but on a larger scale. The median time to surgery after initiating neoadjuvant therapy was 5 weeks, and at the end of the 13-month follow-up period, none of the patients were found to have recurrence. Grade 3–4 treatment-related adverse events were only seen in 3% of the cohort, suggesting that a short and directed course of neoadjuvant ICIs may limit toxicity and is fairly well-tolerated.

Medically inoperable metastatic colon cancer is devastating news for a patient to receive, but ICIs have offered patients with dMMR metastatic colon cancer a chance at remission. The previously mentioned Phase III KEYNOTE_177 study evaluated Pembrolizumab against chemotherapy, specifically in dMMR metastatic CRC, with a higher complete pathologic response rate and a near doubling of progression-free survival time for Pembrolizumab compared to the standard chemotherapy arm [22]. Almost 50% of patients were progression-free at 24 months and were able to discontinue treatment [22]. On the other hand, twice as many patients in the Pembrolizumab arm were found to have disease progression at the first follow up compared to the patients in the chemotherapy arm (29.4% vs. 12.3%) [22]. This has been attributed to three possible causes: primary resistance, misdiagnosis of dMMR status, and pseudoprogression [36], which is defined as progression following initiation of therapy in the form of increased tumor size or presence of new lesions, followed by an overall decrease in tumor burden [37]. Interestingly, despite the initial increased disease progression in the Pembrolizumab arm compared to the chemotherapy arm, follow up at a median of 44.5 months for the KEYNOTE_177 study demonstrated an improved survival for Pembrolizumab, although this improvement was not statistically significant [38], suggesting that more cases of progression may be attributed to pseudoprogression than initially thought. Therefore, the ability to identify pseudoprogression is extremely important, such that patients are not prematurely taken off of ICI therapy. Pseudoprogression can be identified using tumor response scores, such as immune Response Evaluation Criteria in Solid Tumors (iRECIST) [39] and immune-modified Response Evaluation Criteria in Solid Tumors (imRECIST) [40], along with biopsies of new or enlarging lesions and radiographic follow up [37].

Similarly, Lenz et al. recently published the results of their Phase II CheckMate 142 trial regarding patients with metastatic CRC who were treated with combination Nivolumab and low-dose Ipilimumab as a first-line therapy [41]. Of this cohort, 13% achieved a complete pathologic response with a response rate of 69%, with a median follow up of nearly 2 years [41]. Pembrolizumab is often the preferred choice of therapy, but these results have allowed Nivolumab with or without Ipilimumab to become an alternative option for patients with dMMR colon cancer who have an intolerance or contraindication to Pembrolizumab.

### 2.4. ICIs in dMMR Rectal Cancer

Because dMMR colon cancer responded so well to PD-1 inhibitors, it has been hypothesized that rectal cancer demonstrating these mutations may also respond similarly. Cercek et al. performed a very recent and remarkable prospective phase II study evaluating the PD-1 inhibitor Dostarlimab in patients with dMMR rectal cancer. A total of 12 patients with stage II or III rectal adenocarcinoma underwent a Dostarlimab regimen consisting of a 6-month duration with treatment every 3 weeks. All 12 patients had a complete clinical response, which was sustained through at least 6 months of follow up [42]. Adverse effects were lower than grade 3 across all patients [42]. The authors concluded that further follow up would be needed to determine the duration of response, but as with Pembrolizumab, if this drug can induce a durable clinical response, it may become the standard of care for dMMR rectal cancer. Currently, Pembrolizumab and the Nivolumab/Ipilimumab combinations are only approved for use in metastatic rectal cancer. However, much like with the NICHE-2 trial, the results from the study by Cercek et al. have laid the groundwork for a promising future of ICI use in the neoadjuvant setting and will spur further much needed research in this field.

### 2.5. ICIs in pMMR Colon Cancer

Because only a small number of patients have dMMR/MSI CRC, there has been a push for finding immunotherapy regimens that will work for those with pMMR/MSS cancer. Regimens that worked for dMMR CRC have been tested in pMMR CRC, which accounts for the vast majority of CRC, but have had disappointing results. This is attributed largely to two mechanisms of resistance: immunoexclusion and lack of antigens [43].

Immunoexclusion (Figure 1) is based on the principle that tumors can be divided into spatial compartments, including the core and invasive margin, which have varying densities of T cells. Tumors are considered “hot” if there is a high density of T cells in the core, “excluded” if there is high density only at the outer invasive margin and not the core, and “cold” if all areas of the tumor have low T-cell density [44]. Examination of pMMR CRC tumors show that they fall under the “excluded” category, and this lack of T cells at the tumor core may explain, in part, why ICI monotherapy, which requires direct contact with T cells, has been unsuccessful [45]. Additionally, pMMR CRC has a lower mutational burden and fewer antigens that can be detected by cytotoxic T cells, which are associated with a poor response to ICIs [43,46].

Because of this, the focus has shifted from monotherapy to evaluating combinations of ICIs with other therapies. The IMblaze370 trial, for example, was a multi-center and open-label phase 3 randomized controlled trial that evaluated Atezolizumab, an anti-PD-L1 monoclonal antibody, alone and in combination with Cobimetinib, a MEK inhibitor [47]. These interventions were compared against monotherapy use of Regorafenib, an oral multikinase inhibitor that has been shown to have a marginal survival benefit in tumors that progressed on first- and second-line therapy [48]. Most of the patients enrolled had MSS CRC, with a 5% cap on MSI-H CRC patient recruitment. Interestingly, while safety of the Atezolizumab and Cobimetinib combination was consistent with that of the two drugs individually, the combination did not demonstrate improved overall survival [47]. On the other hand, there have been some studies that demonstrate encouraging responses to combination therapy. A study by Fukuoka et al., for example, evaluated the combination of Regorafenib with Nivolimumab in patients with advanced gastric cancer and CRC. This combination had an objective response rate of 33.3% in MSS CRC patients [49], suggesting that further investigations into ICI combination therapy are still warranted, despite the negative results from the IMblaze370 trial. Chemotherapy and radiation have been proposed to induce tumor sensitization to ICIs by various mechanisms, including converting a cold or excluded tumor to a hot one by recruiting T cells to the tumor microenvironment [50] and inducing PD-L1 expression in the cancer cells as a possible target for immunotherapy [51,52].

There will likely be a future role for ICIs in treating pMMR/MSS CRC, but mostly in combination with other drugs and interventions, all of which would aim to sensitize the tumor to immunotherapy. There are various ongoing clinical trials examining the efficacy and safety of ICI combination therapy that will be discussed in the next section. Additionally, there are studies indicating that the microbiome [53,54] and diet [55] can alter tumor responses to ICIs. These possible therapeutic avenues should also be further explored in the context of pMMR/MSS CRC.

### 2.6. Clinical Trials

There are numerous ongoing clinical trials that seek to evaluate combinations of chemotherapy and/or targeted therapies with different ICIs (Table 1). Pembrolizumab, with its success in MSI CRC, is now being tested in combination with Oxaliplatin and Pemetrexed in chemotherapy-resistant metastatic CRC (NCT03626922), as well as with Xelox and Bevacizumab in MSS cancers with high immune infiltrates under the hypothesis that this may make the tumor more sensitive to immunotherapy (NCT04262687). Nivolumab is currently being tested in multiple clinical trials, including for MSS CRC in combination with Encorafenib and Binietinib (NCT04044430), for MSS CRC in combination with the PI3Kinase inhibitor Copanlisib (NCT03711058), and in refractory metastatic MSS CRC, combined with the Epidermal Growth Factor Receptor (EGFR) competitive inhibitor Panitumumab and Ipilimumab (NCT03442569). Atezolizumab is a PD-L1 inhibitor mostly used as an adjuvant therapy in non-small-cell lung carcinoma that is being tested in refractory metastatic CRC in combination with Capecitabine and Bevacizumab (NCT02873195). Avelumab is a PD-L1 inhibitor that is currently approved for use in metastatic Merkel cell carcinoma and urothelial carcinoma that is being tested in combination with Cetuximab and Irinotecan for metastatic MSS CRC (NCT03608046). Avelumab was also evaluated in the recently completed AVETUX trial, which confirmed that adding Avelumab to standard CRC treatment is safe. The authors of the AVETUX trial also identified a certain subpopulation of patients, specifically those with the *FcγR3a* V-allele [56], that showed a trend towards clinical benefit in otherwise unsuccessful drug combinations, concluding that further research into these subpopulations may yield interesting results.

There are several studies evaluating the combination of radiation therapy with ICIs as well (NCT04109755, NCT04017455, NCT03104439). Radiation therapy is used more commonly in rectal cancer and infrequently in colon cancer, but can be beneficial in the treatment of distant lesions in MSS metastatic colon cancer. Interestingly, some of these studies have shown that certain patients demonstrate an abscopal response, defined as a response in non-irradiated distant tumors after receiving the combination therapy of radiation to a targeted tumor and ICIs [57,58].

Lastly, there are also many novel clinically designed antibodies and fusion proteins that are being studied as monotherapy or in combination with other modalities of systemic treatment. For example, TQB2450 is a novel humanized anti-PD-L1 monoclonal antibody that has shown promising results in the treatment of multiple tumors [59] and is currently being studied in combination with immunotherapy, chemotherapy, and other novel agents (NCT05645315, NCT04611711, NCT05139082, NCT03897283). AK104, also known as Cadonilimab, is a bispecific antibody targeting both PD-1 and CTLA-4. Clinical trials evaluating AK104 aim to determine its efficacy in PD-1/PD-L1 blockade-resistant MSI-H/dMMR CRC (NCT05426005), as a perioperative treatment for locally advanced MSI-H/dMMR CRC (NCT04556253), and in combination with mFOLFOXIRI in locally advanced CRC (NCT00571644). There are many other drugs, including ASC61 (an oral PD-L1 inhibitor), SI-B003 (another bispecific anti-PD-1/CTLA-4 antibody), ONC-392 (humanizedanti-CTLA-4 IgG1 monoclonal antibody), and JK08 (IL15 antibody fusion protein targeting CTLA-4), that are detailed in the table below.

### 2.7. Recent Advancements in Molecular Mechanisms of PD-1/PD-L1 Blockade

Although the overall effect of PD-1/PD-L1 blockade in tumors amenable to this class of immunotherapy is upregulation of antitumor activity, the exact mechanisms of this are still under investigation. It is understood that drugs targeting the PD-1/PD-L1 pathway exert their influence by re-invigorating exhausted T cells within the tumor microenvironment [60], but recent studies indicate that there may be many more interactions at play here than first initially realized. Tumors that do not express surface PD-L1 have been shown to demonstrate response to anti-PD-L1 therapy [61,62], and it has been suggested that the presence of PD-L1 on myeloid cells in the tumor microenvironment plays an essential role. Treatment of macrophages with anti-PD-L1 antibodies in both mouse and human models has demonstrated a resulting increase in spontaneous macrophage activation, proliferation, and survival, along with increased tumor infiltration of these macrophages [63], consistent with this theory.

There have also been studies looking at the effects of PD-1/PD-L1 blockade on dendritic cells (DCs) and natural killer (NK) cells, both within and outside of the tumor microenvironment, particularly in tumor-draining lymph nodes. DCs within the tumor microenvironment have been shown to demonstrate in cis binding of PD-L1 to B7.1, which usually binds CD28 downstream and enhances T-cell priming [64]. This in cis binding sequesters B7.1, but treatment with anti-PD-L1 antibodies prevents B7.1 sequestration, allowing DCs in the tumor microenvironment to exert their T-cell priming effects [64]. DCs in the tumor-draining lymph nodes have been shown to upregulate PD-L1 expression upon ingestion of apoptotic tumor cells, which similarly results in decreased T-cell induction [65], but when PD-L1 is deleted in DCs, tumor growth is restricted as a result, and antitumor CD8^+^ T-cell responses are enhanced [66]. These findings suggest that manipulation of dendritic cell PD-L1 expression may be a good target for improving responses to immunotherapy. Similarly, NK cells in the tumor-draining lymph nodes also express surface PD-L1 and have been shown to have improved antitumor effects in leukemia [62] and hepatocellular carcinoma [67] when exposed to a combination of cell-activating cytokines and anti-PD-L1 antibodies. This likely occurs because blockade of surface PD-L1 in NK cells preserves the antitumor benefits of NK cells without allowing them to inactivate intratumoral PD-1^+^ T cells [67].

PD-L1 may also be secreted by tumors in exosomes, which contributes to both local and systemic resistance to the body’s natural antitumor response by stimulating macrophages and monocytes to display a protumorigenic phenotype [68]. Patients demonstrating high levels of exosomal PD-L1 are thought to be resistant to immunotherapy, although the findings in the current literature are conflicting and inconclusive [68]. Further research is needed before exosomal PD-L1 can be used as a predictive biomarker for response to therapy.

While most antibodies target surface or extracellular PD-1/PD-L1, there is also a role for intracellular PD-L1 that has recently been discovered. Intracellular PD-L1 binds to RNA targets and protects them from degradation, thereby promoting resistance to DNA-damaging therapies, such as radiation and chemotherapy [69]. Tu et al. developed an anti-PD-L1 antibody, H1A, that targets intracellular PD-L1, allowing for destabilization of the PD-L1 and RNA complex, which could allow for improved sensitization to other systemic therapies, such as radiation and chemotherapy [69].

Lastly, photodynamic immunotherapy is a newer immunotherapy modality that functions by applying light irradiation to photosensitizers, inducing reactive oxygen species that are then taken up by tumor cells, leading to apoptosis [70]. There have been several studies demonstrating that overexpression of PD-L1 in the tumor microenvironment can also hamper the effect of photodynamic immunotherapy [71,72]. In CRC, nanoparticles have been used to improve delivery of photodynamic immunotherapy to tumors that otherwise would have poor exposure to light irradiation [70]. These nanoparticles have also been shown to decrease the growth of solid tumors [73] and disrupt the PD-1/PD-L1 cascade, resulting in an improved antitumor T-cell effect. Further discussion of photodynamic immunotherapy and nanoparticles is beyond the scope of this paper, but this does represent a newer and fertile area of research that should be explored.

## 3. Adoptive Cell Transfer Therapy

Adoptive cell transfer (ACT) therapy is a type of immunotherapy that revolves around the recognition of neoantigens by immune cells and the transfer of those cells into patients, thereby improving the cancer-fighting abilities of the body’s own immune system.

Immune cells used in ACT therapy may be taken from patients themselves (autologous), may be from donors (allergenic), or may be derived from stem cells. These cells have often undergone gene modification to make them more effective at recognizing a certain neoantigen and eliminating the cell expressing it [16].

There are two main classes of ACT therapy: cytokine-induced killer cells (CIKs) and chimera antigen receptor (CAR) cell therapy [74].

### 3.1. Cytokine-Induced Killer Cells

CIKs are lymphocytes created by culturing autologous peripheral blood mononuclear cells in the presence of interferon-gamma, anti-CD3 monoclonal antibodies, and interleukin 2. These cells are then returned to the patient where they exert their cytotoxic functions against cancer cells [75].

To date, there have been multiple studies evaluating the efficacy and safety of CIK therapy in CRC (Table 2), although it has yet to become a mainstay of treatment. One of the obstacles to this is the large quantity of cell transfusions that need to occur due to poor cell-migration ability [76]. Nonetheless, there are multiple studies showing that the side-effect profile of CIK therapy is tolerable [77,78], consisting most frequently of mild fevers, headaches, chills, and fatigue [79]. Patients who received CIK therapy also report an improved quality of life [78,80]. There are ongoing clinical trials that aim to further evaluate CIK therapy as an option in different types of solid-organ tumors, including CRC, liver cancer, and kidney cancer (NCT04476641, NCT04282044).

### 3.2. Chimera Antigen Receptor Therapy

CAR therapy is a form of immunotherapy produced by separating the patient’s own leukocytes from samples of their whole blood, modifying them to express chimeric receptors on the cell surface, and then re-introducing them into the patient where they can elicit an immune response against any cancer cells expressing the targeted antigens [81]. Despite the possible benefits that CAR therapy can provide, there are a number of unique adverse effects that have also been documented, including cytokine release syndrome, neurologic toxicity and encephalopathy, and various types of cytopenia, leading to infection [82]. It is important for providers to be aware of these adverse effects, especially in patients with risk factors, such as high tumor burden, so that the appropriate interventions can be started should these adverse effects occur.

This type of therapy has seen the most success against hematological malignancies in the United States [83]. In solid tumors, however, there is a lack of specific antigens available for cells to recognize. Therefore, the majority of current research in solid tumors focuses on discovering which specific antigens would make the best targets (Table 2).

Carcinoembryonic antigen (CEA) has been evaluated as a target due to its overexpression in at least half of all CRC [74]. CEA is usually not detected in most normal adult tissues, which may contribute to the overall tolerance to treatment seen in studies evaluating the safety profile of CEA CAR T-cell therapy [84]. There are several studies seeking to further evaluate the efficacy of CEA CAR T cells in CRC (NCT04513431, NCT05415475, NCT05396300), including specifically in refractory or relapsed cancer (NCT04348643) and metastatic disease (NCT05240950).

The epithelial-cell adhesion molecule (EpCAM) is another target of interest. It is a transmembrane cell-adhesion molecule that is expressed by cancers of epithelial origin, such as breast cancer, head and neck cancers, and gastrointestinal cancers, including CRC. Its presence has been associated with cell invasion and migration [85]. It is also one of the most commonly used surface markers to detect cancer cells in various parts of the body, for instance, in the blood. Because of this, trials looking at EpCAM CAR T-cell therapy are still evaluating its efficacy in a variety of different solid-tumor malignancies, including CRC (NCT05028933). Other targets of CAR T-cell therapy under investigation include HER2 (NCT03740256) and c-met (NCT03638206).

One hypothesis regarding the lack of significant response to CAR T-cell therapy in solid tumors examines the barriers that exist around them, preventing infiltration of the T cells to the tumor core. These include physical barriers, such as tumor endothelium [83] and stroma [81], metabolic barriers, such as increased oxidative stress from increased lactic acid production within tumors [86], and immunologic barriers, such as the presence of certain cytokines within the tumor microenvironment that inhibit the proliferation and function of CAR T cells [87]. Certain strategies have been proposed to counteract these barriers, such as direct administration to the tumor instead of systemic administration [88], as well as co-administration with various substances, such as collagenases that could help break down physical barriers [89]. Further research is still needed in these areas. 

### 3.3. Combinations

As with other types of immunotherapy, there has been some investigation into whether combinations of ACT therapy with other types of systemic therapy would improve the overall efficacy (Table 2). For example, under the premise that CIK therapy confers a short-term anti-cancer effect, which is helpful against the primary tumor but may not be long-lasting enough to affect tumor relapse, Gao et al. evaluated whether a combination of CIK therapy and the dendritic cell (DC) vaccine would improve this. The DC vaccine is known to induce tumor-specific memory T cells, which confers a longer-lasting effect. Subjects were randomized to control and experimental groups. Experimental subjects were given DC and CIK therapy while controls were not. Analysis revealed a statistically significant decrease in the risk of tumor progression as well as a statistically significant increase in overall survival [90].

Similarly, there are many current clinical trials that seek to evaluate ACT therapy with other combinations, including standard chemotherapy (NCT03950154, NCT03904537) and Bevacizumab (NCT03950154, NCT02487992). T cells that have undergone genetic editing to increase their efficacy by inhibiting negative regulators of T cell function are also tested in combination with chemotherapy (NCT04426669). Lastly, there are also a few genetically modified NK and DC therapies that are undergoing evaluation, which are further detailed in Table 2.

## 4. Tumor Vaccines

Tumor vaccines are a form of treatment aimed at preventing cancer relapse and recurrence by exposing immune cells to cancer neoantigens [74]. Multiple different types of vaccines exist, including peptide-based, cell-based, such as dendritic cell vaccines, viral vector-based, including oncolytic viruses, and nucleic acid-based vaccines. Unfortunately, the vaccines have not been particularly efficacious, and there has not been significant success in utilizing them in patient care [91].

Peptide vaccines present specific neoantigens to the immune system to generate a memory T cell response against cancers expressing the presented neoantigen. Many of the current trials (Table 3) in CRC seek to study neoantigen vaccines in combination with ICIs (NCT05141721, NCT05243862, NCT05350501, NCT02600949) and chemotherapy (NCT05130060). There is also a study that examined the effects of a neoantigen vaccine combined with ICIs in various types of solid-organ tumors, specifically including MSS CRC (NCT03953235).

DNA vaccines, which have been transferred by vectors, such as bacteria, yeast, and viruses, involve antigen-presenting cells (APCs) and induce a long-lasting humoral and cellular response. Another type of vaccine is the mRNA vaccine, which encodes blueprints for specific neoantigens. After administration, APCs then translate the mRNA to a neoantigen peptide and present it on MHC molecules, stimulating the cytotoxic T-lymphocyte and memory T cell immune response against tumor cells that express the same neoantigen [92]. There is one recruiting clinical trial that is evaluating a double-stranded RNA vaccine in combination with ICIs (NCT04799431).

Dendritic cell (DC) vaccines are most frequently created by pulsing DCs with autologous tumor lysates, thus, exposing the DCs to the tumor antigens unique to the patient’s tumor [93]. This method only works if enough immunogenic tumor is able to be retrieved during resection. In situations where there is an insufficient amount of immunogenic tumor retrieved, DCs can also be exposed to more generalized antigens, such as CEA, although this has not been shown to induce a significant survival benefit in most patients [94,95,96]. One of the reasons for this lack of success is likely the inability of DC vaccines to circumvent the immunosuppressive aspects of the tumor microenvironment [93]. It has been hypothesized that combinations of DC vaccines with other immunosuppressive therapies, such as ICIs and anti-angiogenic medications, could improve the efficacy. There are several recruiting clinical trials examining DC vaccines in CRC at this time, with (NCT04912765, NCT02919644) and without (NCT03730948) co-administration of other drugs.

Oncolytic virus vaccines can directly induce tumor cell death through lysis or through stimulation of immunogenic cell death. Additionally, infection of the tumor by these oncolytic viruses has been shown to change various cold tumors, including brain, colorectal, and breast tumors, into hot tumors, which has the potential to greatly increase the efficacy of concurrent ICI administration [97,98]. As such, the oncolytic viral vaccine is another therapy that should be studied as a possible immunomodulator to improve response to ICIs in MSS CRC. There are also some other interesting viral-based vaccine trials currently ongoing. One looks at the possible neoadjuvant use of intratumoral influenza vaccine in early CRC to downstage and enhance the immune response against the tumor (NCT04591379). Another evaluates a poxvirus vaccine that encodes the two specific tumor antigens, CEA and mucin-1, in small bowel cancer and MSS CRC (NCT04491955).

Yet another vaccine that has been tested in the setting of MSS CRC is Colon GVAX, which is an allogenic whole-tumor cell vaccine that has been genetically modified to produce immune stimulatory cytokines, such as granulocyte–macrophage colony-stimulating factor. This increases T cell immunity against the tumor-associated antigens presented by the whole tumor cells [99]. One study evaluated the effects of GVAX being co-administered with Pembrolizumab and Cyclophosphamide in 17 patients with pMMR CRC. The authors concluded that GVAX may modulate the response to immunotherapy given the biochemical changes that were observed in the subset of patients who had previously had no response to Pembrolizumab monotherapy. GVAX continues to be used in ongoing clinical trials, specifically looking at its effect in metastatic CRC (NCT01952730), and should also be further studied to for its potential to sensitize MSS CRC to ICIs.

## 5. Conclusions

Immunotherapy is a new therapeutic modality with proven benefits in the MSI CRC population. There are numerous ongoing clinical trials looking to find the next big breakthrough that will allow immunotherapy to play an even bigger role in the treatment of CRC. Recent advances have pointed to a significant role for ICIs in the neoadjuvant setting for both dMMR colon and dMMR rectal cancer, which would be a novel application of an existing drug class. The use of ACT and tumor vaccines has not yet found widespread use, but once efficacy and safety profile can be improved, these therapies could greatly increase our ability to control advanced tumors and tumor relapse or recurrence. Particular attention should be paid to the potential of developing immunotherapy that would treat the MSS subset of CRC, which makes up the majority of all CRC cases.

## Figures and Tables

**Figure 1 cells-12-00258-f001:**
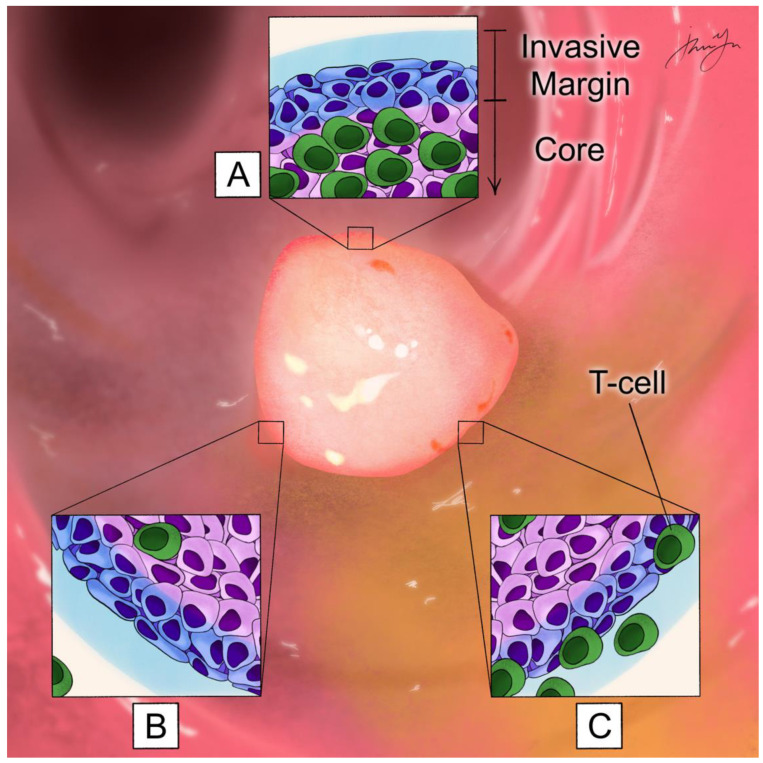
The principle of immunoexclusion is based on the concept that tumors can be hot, cold, or excluded based on the location and density of T cells in the tumor core and/or invasive margin of the tumor, which is defined as a 1 mm region centered on the interface of malignant cells and normal host cells. (**A**). Hot tumors have high T-cell density in the core of the tumor. (**B**). Cold tumors have low T-cell density in the core and invasive margin of the tumor. (**C**). Excluded tumors have high T-cell density only in the invasive margin of the tumor.

**Table 1 cells-12-00258-t001:** Clinical trials evaluating immune checkpoint inhibitors (ICIs) in combination with chemotherapy, radiation, and/or other ICIs for colorectal cancer (CRC). Included also are trials evaluating clinically designed antibodies and fusion proteins in CRC.

Clinical Trial	Intervention	Target Population	Primary Objectives
NCT03626922	Pembrolizumab Oxaliplatin Pemetrexed	Chemotherapy-resistant and metastatic microsatellite stable (MSS) CRC	Establish dose Determine efficacy
NCT04262687	Pembrolizumab Xelox Bevacizumab	MSS CRC with high immune filtrate	Progression-free survival (PFS) at 10 months
NCT04044430	Nivolumab Encorafenoib Binietinib	Metastatic MSS, BRAFV600E gene-mutated CRC	Overall response rate (ORR) Determine safety and tolerability
NCT03711058	Nivolumab Copanlisib	Refractory/relapsed MSS CRC	Maximum tolerated dose (MTD) of Copanlisib ORR
NCT03442569	Nivolumab Panitumumab Ipilimumab	Unresectable, refractory metastatic KRAS/NRAS/BRAF wild-type MSS CRC	ORR
NCT02873195	Atezolizumab Capecitabine Bevacizumab	Metastatic, recurrent, and/or refractory CRC	PFS up to 20 months
NCT03608046	Avelumab Cetuximab Irinotecan	Metastatic MSS CRC	ORR
NCT05645315	TQB2450 (PD-L1 monoclonal antibody) TQB2618 (TIM-3 receptor monoclonal antibody)	Advanced malignant solid tumors	Dose-limiting toxicity (DLT) Phase II recommended dose (RP2D) ORR
NCT04611711	TQB2450 Decitabine Anlotinib	PD-1 monoclonal antibody-resistant digestive system tumors	ORR
NCT05139082	TQB2450 TQB3616 (CKD4/6 inhibitor)	PD-1/PD-L1 monoclonal antibody-resistant digestive system tumors	ORR
NCT03897283	TQB2450 Anlotinib	Advanced solid tumors	DLT MTD RP2D
NCT05287399	ASC61 (orally bioavailable small-molecule PD-L1 inhibitor)	Advanced solid tumors	Proportion of patients experiencing DLT RP2D
NCT04109755	Neoadjuvant 5Gy x 5 doses + Pembrolizumab	MSS rectal cancer	Tumor regression rate
NCT04017455	Neoadjuvant radiation + Bevacizumab and Atezolizumab	Resectable rectal cancer	Clinical complete and near-complete response rate
NCT03104439	Radiation + Nivolumab and Ipilimumab	MSS CRC, high microsatellite instability (MSI-H) CRC	Disease control rate
NCT05187338	Ipilimumab Pembrolizumab Durvalumab	Advanced solid tumors including CRC	Safety PFS at 5 years Disease control rate Duration of remission
NCT03539822	Cabozantinib Durvalumab Tremelimumab	Gastrointestinal tumors including CRC	MTDORR
NCT04606472	SI-B003 (bispecific anti-PD-1/CTLA-4 antibody)	Advanced solid tumors including CRC	DLT MTD Maximum administered dose Treatment-related AEs RP2D
NCT05426005	AK104 (bispecific anti-PD-1/CTLA-4 antibody)	PD-1/PD-L1 blockade-refractory MSI-H or mismatch repair deficient (dMMR) advanced CRC	ORR
NCT04556253	AK104	Locally advanced MSI-H/dMMR gastric carcinoma and CRC during perioperative period	Complete pathologic response (pCR) rates
NCT00571644	AK104 mFOLFOXIRI	Locally advanced CRC	pCR rates
NCT00571293	Neoadjuvant Balstilimab with Botensilimab	Resectable CRC	Pathological overall response Number of adverse events (AEs) Number of serious AEs Number of complications leading to operative delay
NCT05627635	Balstilimab Botensilimab FOLFOX Bevacizumab	MSS metastatic CRC	Safety and tolerability RP2DORR
NCT05608044	Balstilimab Botensilimab	Metastatic CRC	ORR
NCT03860272	Balstilimab Botensilimab	Refractory solid tumors including CRC without hepatic metastases	Incidence of AEs DLT of Botensilimab RP2D of Botensilimab
NCT04140526	ONC-392 (humanized anti-CTLA-4 IgG1 monoclonal antibody) Pembrolizumab	Advanced or metastatic solid tumors including CRC	DLT in monotherapy MTD in monotherapy RP2D Rate of AEs
NCT05620134	JK08 (IL-15 antibody fusion protein targeting CTLA-4)	Unresectable locally advanced or metastatic CRC	DLT RP2D Safety and tolerability including nature of AEs

**Table 2 cells-12-00258-t002:** Clinical trials evaluating the use of adoptive cell transfer therapy (ACT) in CRC.

Clinical Trial	Intervention	Target Population	Primary Objectives
NCT04476641	Dendritic cell (DC) activated cytokine-induced killer cell (CIK)	Solid tumors including CRC	OS and PFS
NCT04282044	Autologous CIK	Advanced solid tumors including CRC	AEs DLT
NCT04513431	Carcinoembryonic antigen (CEA) chimera antigen receptor (CAR) T cells	CEA-positive stage III CRC CRC liver metastases	Treatment-related AEs
NCT05415475	CEA CAR T cells	CEA-positive advanced solid tumors including CRC	Treatment-related AEs MTD of CEA CAR T cells
NCT05396300	CEA CAR T cells	CEA-positive advanced solid tumors including CRC	Treatment-related AEs MTD of CEA CAR T cells
NCT04348643	CEA CAR T cells	Refractory/relapsed CEA-positive CRC	Treatment-related AEs
NCT05240950	CEA CAR T cells	Metastatic CRC after adjuvant chemotherapy	Treatment-related AEs Recurrence Recurrence-free survival
NCT05028933	EpCAM CAR T cell	Advanced digestive system malignancies including CRC	Evaluate safety and tolerability
NCT03740256	HER2 CAR T-cell Oncolytic adenovirus	HER2-positive solid tumors including CRC	Incidence of DLT
NCT03638206	c-met CAR T cell	Various malignancies including CRC	Treatment-related AEs
NCT03950154	PD-1-T lymphocytes Xelox Bevacizumab	Recurrent and metastatic CRC	PFS
NCT03904537	Anti-PD-1 tumor-infiltrating lymphocytes (TILs) Xelox	Stage III CRC undergoing adjuvant chemotherapy	Disease-free survival
NCT03950154	PD-1-T lymphocytes Bevacizumab Xelox	CRC	PFS
NCT02487992	CIK plus S-1 Bevacizumab	Advanced CRC	OS
NCT04426669	Cytokine inducible SH2 containing protein (CISH)-inactivated TILs Cyclophosphamide Fludarabine Aldesleukin	Gastrointestinal cancers including CRC	MTD Changes in tumor dimension Treatment-related AEs
NCT04842812	Anti-PD-1 and anti-CTLA-4 CAR-TILs	Advanced solid tumors including CRC	Safety of treatment
NCT05213195	Natural killer group 2 member D (NKG2D) CAR natural killer (NK) cell therapy	Refractory metastatic CRC	DLT MTD
NCT05248048	NKG2D CAR-T cell therapy	Previously treated CRC with liver metastases	DLT MTD
NCT05194709	Anti-oncofetal trophoblast glycoprotein (5T4) CAR-NK cells	Advanced solid tumors	Incidence of AEs
NCT05631886	TP53-EphA-2-CAR-DC PD-1 antibodies Abraxane Cyclophosphamide	Adult malignant solid tumors	Incidence of AEs Clinical response Immune response
NCT05631899	KRAS-EphA-2-CAR-DC PD-1 antibodies Abraxane Cyclophosphamide	Locally advanced or metastatic solid tumors	Incidence of AEs Clinical response Immune response

**Table 3 cells-12-00258-t003:** Clinical trials evaluating the use of tumor vaccines in CRC.

Clinical Trial	Intervention	Target Population	Primary Objectives
NCT05141721	Patient-specific neoantigen vaccine Atezolizumab Ipilimumab Fluoropyrimidine Bevacizumab	Metastatic CRC	Molecular response
NCT05243862	PolyPEPI1018 vaccine Atezolizumab	Relapsed/refractory MSS CRC	Treatment-related adverse events Administration safety
NCT05350501	EO2040 vaccine Nivolumab	Minimal residual disease in Stage II, III, or IV CRC after curative therapy	Response to treatment at 6 months
NCT02600949	Personalized peptide vaccine Pembrolizumab Sotigalimab	Advanced or metastatic CRC or pancreatic cancer	Demonstrate vaccine feasibility Establish vaccine safety
NCT04117087	KRAS peptide vaccine Nivolumab Ipilimumab	Resected mismatch-repair proficient (pMMR) CRC and pancreatic cancer	Number of drug-related toxicities Fold change in interferon-producing mutant-KRAS-specific CD8 and CD4 T cells
NCT05130060	PolyPEPI1018 vaccine TAS-102	Metastatic CRC	Evaluate safety and tolerability
NCT03953235	GRT-C903/GRT-R904 vaccine Nivolumab Ipilimumab	Solid organ tumors Including MSS CRC	Treatment-related adverse events ORR Establish Phase II dose
NCT04799431	Neoantigen vaccine and Poly-ICLC Retifanlimab	Stage IV MSS CRC and pancreatic ductal adenocarcinoma	Percentage receiving vaccine Treatment-related adverse events
NCT04912765	Dendritic cell (DC) vaccine Nivolumab	Liver metastases from CRC and HCC	Relapse-free survival at 24 months Induced immune response
NCT02919644	DC vaccine IL2	Stage IV CRC after curative resection	Treatment-related adverse events Immunological efficacy
NCT03730948	mDC3 vaccine Cyclophosphamide	Resected hypermutated CRC	Change in number of peptide-specific CD8^+^ T cells Treatment-related adverse events
NCT04591379	Neoadjuvant intratumoral influenza vaccine	CRC	Evaluate safety
NCT04491955	Pox-virus vaccine against CEA & mucin-1	MSS CRC, small bowel cancer	ORR
NCT01952730	GVAX	Stage IV CRC	Treatment-limiting toxicity

## Data Availability

Not applicable.

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
