# Peer review of "Immunotherapy: Recent Advances and Its Future as a Neoadjuvant, Adjuvant, and Primary Treatment in Colorectal Cancer"

_cells, 2023, doi:10.3390/cells12020258_

Round 1
Reviewer 1 Report (New Reviewer)
In this research, the authors reviewed the “Immunotherapy: Recent Advances and its Future as a Neoad-juvant, Adjuvant, and Primary Treatment in Colorectal Cancer” In my opinion, the current stage of this paper could meet the requirements of Cells after major revisions.
My comments are as details:
1. In the part of “History of Immune Checkpoint Inhibitors”. The authors only reviewed the PD-1, PD-L1, or CTLA-4 pathway simply. In my opinion, more information should be discussed, including the function or mechanism of these proteins, how it was regulated, and the currently used small molecular inhibitors. Some related researches may be helpful to the authors: 10.1016/j.molcel.2019.04.005; 10.1002/adma.202206121; doi.org/10.1016/j.apsb.2022.07.023; doi.org/10.1016/j.jconrel.2022.11.004; 10.1126/scitranslmed.aav7431.
2. Some minor mistakes exist in the paper: Line 73, CD4+ or CD8+ T-cells should be CD4+ or CD8+ T-cells. The authors should carefully check it.
3. In Table 1, Some other clinically designed PD-L1, CTLA_4, or PD-1 antibody or small inhibitors should be more clearly summarized, including TQB2450, ASC61, and et.al.
4. In Tab. 2, apart from CAR-T, the current clinical trials of CAR-NK, CAR-DC should be summarized.
5. How PD-L1, PD-1, CTLA-4 affects the function of DCs or macrophages should be revealed. Some research may be helpful to the authors: 10.1038/s43018-020-0075-x; 10.1158/2326-6066.CIR-17-0537; doi/10.1126/scitranslmed.abi4670; 10.1158/2159-8290.CD-18-1259; 10.1016/j.ccell.2020.09.001; 10.1016/j.trecan.2020.03.002;
Author Response
Please see the attachement

Reviewer 2 Report (New Reviewer)
This is an overall well written and informative review about current approaches of immunotherapy for the treatment of colorectal cancer. The review is easy to read and covers the most important aspects in the field.
Minor comments:
- In the chapter on clinical trials of ICI combination treatments, the authors only describe ongoing trials but did not discuss studies with published results such as the Imblaze370 trial. This should be added.
- An ongoing discussion of clinical interest is which diagnostic approach is best to identify dMMR tumors (TMB versus immunohistochemistry versus specific assays). Maybe the authors can discuss this aspect.
Author Response
Please see the attachment

This manuscript is a resubmission of an earlier submission. The following is a list of the peer review reports and author responses from that submission.
Round 1
Reviewer 1 Report
In this manuscript authors have reviewed recent clinical advances in three main classes of immunotherapy - immnune checkpoint inhibitors (ICI), adoptive cell transfer and tumor vaccines in colorectal cancer (CRC) treatment. Immunotherapy is relatively new treatment modality in advanced colorectal cancers. In a landmark clinical trial, a colorectal cancer patient with microsatellite instability (MSI) responded to immune checkpoint inhibitor nivolumab. Deficiency in DNA mismatch repair in cancer cells can cause shortening of length of microsatellite and generation of neoantigens. In another clinical trial anti PD-1 checkpoint pembrolizumab found efficacious in high microsatellite instability mismatch repair deficient CRC. However vast majority of CRC are proficient in mismatch repair (pMMR) were not found to be responsive to similar immune checkpoint inhibitors due to either immunoexclusion or lack of antigen. In this review authors have also summarized clinical trials that seeks to evaluate combinations of chemotherapy/targeted therapy with different ICIs. They have also discussed clinical studies involving adoptive cell transfer therapy such as CAR T cell and tumor vaccines.
Overall article has nicely presented the challenges encountered in field in targeting CRC using immunotherapy and reflected on future strategies needed to overcome these challenges. In my opinion it is scientifically sound and well written manuscript and I recommend it for acceptance for publication.
Reviewer 2 Report
This review of immunotherapy in colorectal cancer provides new insights for current therapeutic clinical trials. The authors are particularly interested in the therapies that would treat the microsatellite stable (MSS) subset of CRC, which makes up the majority of colorectal cancers. They describe several immune therapies such as anti-PD-1, anti-CTLA-4, or their combination, adoptive transfer, CAR T cell therapy, and tumor vaccines. Also, the potential combination with other types of therapies like radiation, EGFR, PI3Kinase inhibitors. I found the information provided by the authors is updated in the field, but I suggest add not only the PROS but also the CONS in the usage of those therapies, reduction of cancer, time, and incidence. Including, if available charts, and a table of the current trials. In general, I think the review is acceptable to publish, with some changes that make clear the type of therapy that is more useful according to the stage, the severity, and potential side effects. The usage of tables to summarize it will be a good idea. Best wishes,Reviewer 3 Report
In the past decade, immunotherapy has attracted increasing attention in the treatment of colorectal cancer due to its favorable oncological effects on specific genotypes of tumors. This review article contained three major topics with an updated introduction for each, especially on the topic of immune checkpoint inhibitors.
There are several suggestions for the munuscript. For immune check blockade, neoadjuvant immune checkpoint inhibition in locally advanced MMR-deficient colon cancer: the NICHE-2 study has recently shown promising oncological effects. It is suggested that immune checkpoint inhibitors for dMMR CRC should be discussed in the phases of neoadjuvant and adjuvant, as well as different stages of CRC, ex early, LACRC, and metastatic CRC. On the topic of tumor vaccine, the reference (56) for oncolytic virus vaccination is not relevant.
In general, the content including the main text, tables, and figures, is not sufficient for the review articles requested on the website of cells. The manuscript's structure is a bit complicated and there are no numbered headings. Please follow the instruction for the authors in cells website https://www.mdpi.com/journal/cells/instructions. By the way, several English grammar problems in terms of misused present and past tense need to be revised.